# Perceived Information Distortion about COVID-19 Vaccination and Addictive Social Media Use among Social Media Users in Hong Kong: The Moderating Roles of Functional Literacy and Critical Literacy

**DOI:** 10.3390/ijerph19148550

**Published:** 2022-07-13

**Authors:** Luyao Xie, Edmund W. J. Lee, Vivian W. I. Fong, Kam-Hei Hui, Meiqi Xin, Pheonix K. H. Mo

**Affiliations:** 1Centre for Health Behaviours Research, JC School of Public Health and Primary Care, The Chinese University of Hong Kong, Hong Kong, China; luyaoxie@link.cuhk.edu.hk (L.X.); vvwi.fong@gmail.com (V.W.I.F.); carloshui@cuhk.edu.hk (K.-H.H.); 2Wee Kim Wee School of Communication and Information, Nanyang Technological University, Singapore 639798, Singapore; edmundlee@ntu.edu.sg; 3Department of Rehabilitation Sciences, The Hong Kong Polytechnic University, Hong Kong, China

**Keywords:** addictive social media use, perceived information distortion, functional literacy, critical literacy, COVID-19

## Abstract

During the COVID-19 pandemic, distorted information about the COVID-19 vaccination is widely disseminated through social media. The present study examined the association between perceived information distortion about COVID-19 vaccination on social media, individuals’ functional and critical literacy, and addictive social media use (SMU), as well as the moderating roles of functional and critical literacy in the association between perceived information distortion and addictive SMU among social media users in Hong Kong. A web-based, cross-sectional survey was conducted among 411 Chinese citizens from June to August 2021. Findings showed that after adjusting for significant background variables, including age, gender, marital status, education, occupation, and income, functional literacy was negatively associated with addictive SMU. In addition, significant moderation effects of functional literacy and critical literacy were also observed, such that a positive association between perceived information distortion on social media and addictive SMU was significant among participants with lower functional literacy or higher critical literacy. Findings highlight the importance of improving functional literacy in addictive SMU prevention for social media users. Special attention should also be paid to the potential influence of critical literacy on addictive SMUs.

## 1. Introduction

COVID-19 has caused severe threats to public health globally, and vaccination is recognized as one of the most effective ways to curb this pandemic’s spread [1]. In Hong Kong, two types of vaccines (i.e., the Comirnaty (BioNTech) mRNA vaccine and the CoronaVac (Sinovac) inactivated vaccine) are currently provided free of charge to the public [2]. Due to the prolonged outbreak, news about the COVID-19 vaccination disseminates widely through social media platforms and constantly becomes a focus of social media discourse [3].

Social media platforms are internet-based applications that enable users to obtain, create, share, and communicate information with others and engage in public discourse [4]. With the popularity of smartphones, social media has become an essential part of people’s daily lives due to its strength in speed in the spread of news and accessibility in the interaction of information in real-time [5,6]. In the face of the global crisis, people actively use social media—some more intensively than others—to access information about COVID-19 vaccines [7]. There are two key concerns about social media use during the pandemic. First, there is a concern about false and misleading news about vaccination, such as low vaccine efficacy, exaggerated side effects, and conspiracy theories, that are widely circulated on social media [8,9,10], leading to a problem known as *information distortion.* Second, as lockdown policies were put in place to curb the pandemic, people spent more time on social media, which may be indicative of a rise in addictive tendencies [11,12]. Currently, there are some gaps in addictive social media use (SMU), and the present study focused on two of them. First, many studies concerning the predictors of addictive SMU focused predominantly on personal factors, such as personality and psychosocial needs [13]. Few have examined the impact of information quality-related factors (e.g., information distortion) on addictive SMU, particularly in the COVID-19 pandemic. Second, electronic health (eHealth) literacy reflects one’s ability to utilize online health information, which is closely associated with one’s perception of information and online behaviors [14]. Yet very few studies have investigated its role in the relationship between information-related factors and addictive SMU. Therefore, this study aimed to fill the gaps highlighted with the following objectives: First, we examined how perceived information distortion regarding COVID-19 vaccination on social media was related to addictive SMU among social media users in Hong Kong. Second, we explored the moderating role of functional literacy in the association between perceived information distortion and addictive SMU. Third, the moderating role of critical literacy in this association was also explored.

### 1.1. Outcome of Interest—Addictive Social Media Use (SMU)

Despite the convenience provided by social media, previous research suggests that intensive online activity may lead to the development of addictive tendencies [15]. During a pandemic, people lean toward excessive use of the Internet and social media, as they are the most accessible and easy-to-use means of information acquisition and interaction [11,12]. Addictive SMU manifests itself as overly concerned about social media, representing the following six typical characteristics: salience, mood, modification, tolerance, withdrawal conflict, and relapse [16]. Addictive SMU can develop into social media addiction (SMA), which refers to individuals exhibiting maladaptive psychological dependency on social media to the extent that addiction-like symptoms occur [17]. SMA can lead to negative consequences, such as unhealthy social relationships [18] and psychological distress [19]. Understanding the factors of addictive SMU will provide practical implications for preventing SMA.

### 1.2. Perceived Information Distortion on Social Media and Addictive SMU

Information distortion refers to the tendency of information to be altered, omitted, or reorganized as it is communicated [20], which is a common phenomenon on social media during the pandemic [10]. It was found that sharing false news with biased and emotionally charged opinions is more prevalent on social media, as it captures more attention than detached, positive, or neutral information [21]. In contrast with traditional media, the content posted on social media need not undergo strict scientific censorship, and social media users can share their personal opinions freely and are thus more susceptible to distorted information [22,23].

Most studies regarding the factors behind addictive or problematic SMU focused predominantly on an individual’s personality (e.g., neuroticism [13], fear of missing out [24]), psychosocial needs (e.g., self-identify [13], need for belongingness and relatedness [25]) and entertainment gratifications [26]. The impact of online information quality on consumers’ addictive SMU has received little attention. During the COVID-19 pandemic, people are in demand of finding reliable and updated information concerning the pandemic and vaccination. Social media has become the primary source for capturing information amid the pandemic [7]. According to the Uses and Gratifications Theory, motivation for SMU is directed by users’ need for gratification [27]. The quality of informational content is closely related to one’s information-seeking motives and informative gratification [28,29]. In other words, individuals’ satisfaction with information quality may predict their behavioral usage intensity. Currently, while social media helps accelerate the dissemination of COVID-19 and vaccine information in a social environment, it also increases the risk of false or distorted information sharing due to a lack of control over the process [30]. Guelmami et al. found that the consumption of social media disinformation related to COVID-19 was positively correlated with social media addiction in the Tunisian population. Individuals’ perceived information distortion on social media might result in extensive online information consumption as perceived information distortion makes it more difficult for consumers to obtain effective information [31]. As a result, they are likely to use social media excessively for more relevant information to reduce the uncertainty caused by the perception of distorted information and meet their informational gratification [29,32], which may result in their addictive SMU tendency due to overuse. As such, we posit the following:

**Hypothesis** **1.**
*Perceived information distortion on social media platforms would be positively associated with one’s addictive SMU.*


### 1.3. Potential Moderators—Functional Literacy and Critical Literacy

Electric health (eHealth) literacy refers to the ability to seek, find, understand, and appraise health information from electronic sources and apply such knowledge gained to address health problems [14], and its importance to individuals’ health has been widely documented [33]. Notably, the importance of eHealth literacy has been emphasized during the COVID-19 pandemic, as it is vital in helping individuals accurately understand and critically evaluate the pandemic- and vaccine-related news and information on social media [34,35]. To date, very few studies have examined the role of eHealth literacy in addictive or problematic SMUs. According to Nutbeam and Chiang et al., functional literacy and critical literacy are two main dimensions of eHealth or health literacy [36,37].

Functional literacy refers to basic skills in reading, understanding, and writing skills of online information [37]. Some studies demonstrated that problematic SMU was common among those with low health literacy or Internet literacy [38,39], both of which contain basic features of functional literacy. Individuals’ perceptions of information distortion on social media and their functional literacy levels are related and may interact with each other [40]. More specifically, individuals with lower functional literacy generally have difficulty fully understanding the content on social media. When they perceive information as being distorted, they may need additional time on social media to bridge the gap by seeking out basic information more intensively for clarification, which may result in addictive SMU. On the other hand, individuals with higher functional literacy can effectively capture and understand such information. Thus, they are less likely to get lost in distorted information or become addicted to social media. As such, we postulate that, as follows:

**Hypothesis** **2.***Gaps in addictive SMU between high and low perceived information distortion would depend on functional literacy*, *such that those with high functional literacy would display lower addictive SMU than those who low functional literacy.*

Apart from functional literacy, we propose that critical literacy would be another potential moderator of the relationship between perceived information distortion and addictive SMU. Critical literacy involves more advanced cognitive skills to critically evaluate online information, discern the quality of platforms, and use quality information to make informed decisions [37]. Previous research indicated that the formation of Internet addiction might be due to the excessive activation of the impulsive system and the ineffectiveness of the reflective system [41]. Individuals with higher critical thinking dispositions can process various kinds of information more rationally through their reflective systems. In other words, they can analyze the whole thing in a reasonable and objective form, which may indicate greater resistance to the impulsive system and thus not being addicted to social media [42]. Therefore, critical literacy, as a particular aspect of critical thinking, may weaken the relationship between perceived information distortion and social media addiction. However, individuals with high critical literacy generally tend to critically evaluate their obtained information, which may increase their perception of information distortion on social media. When perceiving information distortion, people with a higher level of critical literacy may increase their SMU intensity to cope with this situation and discern the quality of information by retrieving and integrating more relevant information or comparing the information from different channels [43]. To meet this quality information gratification, they may also develop addictive SMU tendencies. Due to the mixed findings in existing research, we propose the following research question (RQ):

**RQ1:** 
*How would critical literacy moderate the relationship between perceived information distortion and addictive SMU?*


## 2. Materials and Methods

### 2.1. Participants and Procedure

A web-based, cross-sectional survey was conducted among social media users from June to August 2021 in Hong Kong. The inclusion criteria of the participants were as follows: (1) Hong Kong residents; (2) aged 18 years or over; (3) using social media platforms, such as Twitter, Facebook, Instagram, Snapchat, and TikTok, at least once a week in the past six months; (4) having been exposed to COVID-19 vaccine-related information on any social media platforms in the past month.

Eligible participants were invited to participate in the web-based survey via social networking sites (e.g., WhatsApp, WeChat, etc.). The online survey was accessed through a hyperlink or a QR code for interested participants to log on to Google Forms, including the informed consent and study questionnaire. Participants were informed that the survey was voluntary and confidential, and only those who provided their informed consent could proceed with the following questions. Participants self-administered the online questionnaire, which took 15–20 min to complete. Each participant would receive an HKD20 (about USD2.56) coupon upon completion as a token of appreciation for their participation. Ethical approval was obtained from the ethics committee of the Chinese University of Hong Kong. A total of 411 completed responses were collected.

### 2.2. Measures

*Addictive SMU* was assessed by the six-item Bergen Social Media Addiction Scale (BSMAS) [15], which comprises the following six core components: salience, mood, modification, tolerance, withdrawal conflict, and relapse [16]. The Chinese version of BSMAS has been validated among the Hong Kong population [44], which examines the experience of social media usage over the past three months in this study with 5-point Likert response options from 1 (very rarely) to 5 (very often). Sample items included the following: (a) “How often during the last three months have you spent a lot of time thinking about social media or planned use of social media?” and (b) “How often have you felt an urge to use social media more and more during the last three months?”. Higher scores indicate addictive SMU. The Cronbach’s alpha was 0.880 in the present study. A cut-off score of 19 (out of 30) on this scale for social media addiction (SMA) was proposed by Bányai et al. [45].

*Perceived information distortion about COVID-19 vaccination* (Cronbach’s alpha = 0.849) was measured by two items adapted from a subscale of the media quality scale with vaccine-related context [46]. The items were the following: “Social media exaggerates and sensationalizes negative news about COVID-19 vaccination (e.g., sensationalist reports of adverse events)” and “Social media selectively publishes negative news about COVID-19 vaccination.” The response was rated on a 5-point Likert Scale (1 = strongly disagree to 5 = strongly agree), and a higher score indicated higher perceived information distortion.

*Functional literacy* and *critical literacy* were measured by the functional and critical subscales of the 12-item eHealth literacy scale developed by Hsu et al. [37]. The items were all adapted to the COVID-19 vaccination context in this study. Sample items to measure functional literacy and critical literacy included the following: “I find vaccine information on social media difficult to understand.” and “When I have questions about vaccine information on social media, I verify it through other methods.” The Cronbach’s α of functional literacy and critical literacy in the present study was 0.665 and 0.820, respectively.

### 2.3. Statistical Analysis

Descriptive statistics were first used to present the characteristics of the study population. Next, Pearson correlation analysis was conducted to show the correlations between the key variables in this study. Univariate regression was performed first to examine the associations between background variables and addictive SMU. Then, hierarchical regression was used to examine the association between perceived information distortion and addictive SMU, with functional literacy and critical literacy as potential moderators in such association. All significant background variables in univariate regression were adjusted for in the hierarchical regressions.

Specifically, the main effects of perceived information distortion on social media and functional literacy on addictive SMU were first examined; to test the moderating effect of functional literacy, the two-way interaction term between perceived information distortion and functional literacy was computed and added to the hierarchical regression model. Such analyses were repeated for critical literacy as the second moderator to examine the effect of its interaction with perceived information distortion on addictive SMU. Both unstandardized (B) and standardized coefficients (β) with 95% confidence intervals (CI) in the hierarchical regressions were reported. All the above analyses were conducted using SPSS 26.0, and statistical significance was defined as a *p*-value < 0.05.

## 3. Results

The descriptive statistics are presented in Table 1. The mean age of the respondents was 34.5 (SD = 15.3) years old, and 58.9% were female. A majority of the participants surveyed were single (64%) and had a university or above education level (78.8%). Among all participants, one-third (33.3%) were students, and 44% had full-time employment. Nearly half (45.3%) had a monthly personal income of HKD 10,000 or below. Using the cut-off of 19 (out of 30), about 16.3% of the respondents had social media addiction (SMA).

The pairwise correlations among key variables are presented in Table 2. As shown in Table 2, age, marital status, educational level, monthly income, and functional literacy were significantly associated with addictive SMU at the zero-order level (*p* < 0.05).

Table 3 shows the hierarchical regression analysis of perceived information distortion and functional literacy on addictive SMU. The significant demographic variables in the univariate analysis included age, education, marital status, and monthly income (results not shown in the table). First, Block 1 assessed the relationship between background variables and addictive SMU, and the only significant variable was age (β = −0.19, 95% CI (−0.33, −0.04), *p* < 0.05). Demographic variables explained 5.30% of the variance in addictive SMUs. H1 posited that Perceived information distortion on social media would be positively associated with one’s addictive SMU. Block 2 showed that after adjusting for background variables, perceived information distortion had no significant association with addictive SMU, thus not supporting H1. H2 postulated that gaps in addictive SMU between high and low perceived information distortion would depend on functional literacy. The results in block 2 showed that functional literacy was negatively associated with addictive SMU (β = −0.29, 95% CI (−0.38, 0.20], *p* < 0.001), and block 3 further examined the interaction between perceived information distortion and functional literacy in explaining the variance in addictive SMU with a significantly negative interaction emerged (β = −0.10, 95% CI (−0.19, 0.00), *p* = 0.046). The entire regression block explained 14.80% of the total variance in addictive SMUs. Results of simple slope analysis (Figure 1) showed that the positive association between perceived information distortion and addictive SMU was significant among participants with lower functional literacy (slope b = 0.70, *p* = 0.046) but not significant among those with higher functional literacy. Thus, H2 was supported.

Table 4 presents the results of the effect of critical literacy and its interaction with perceived information distortion in explaining the variance in addictive SMU. Block 2 showed no significant association between critical literacy and addictive SMU after adjusting for background variables. In Block 3, a significant two-way interaction emerged between perceived information distortion and critical literacy (β = 0.10, 95% CI [0.00, 0.19], *p* = 0.04). The entire regression block explained 6.60% of the total variance in addictive SMUs. Results of the analysis of the simple slope (Figure 2) demonstrated that a positive relationship between perceived information distortion and addictive SMU was only significant among the participants with higher critical literacy (slope b = 0.74, p = 0.02), whereas not significant among those with lower critical literacy, which answered RQ1.

## 4. Discussion

This study explored the relationship between the perception of distorted information on social media platforms and users’ addictive SMU tendencies during the COVID-19 pandemic. The most significant aspect of this study is how eHealth literacy moderates the association between an individual’s perceived information distortion on social media and addictive SMU behavior. More specifically, perceived higher information distortion had a stronger effect on addictive SMUs among those with lower functional literacy or higher critical literacy.

Functional literacy and critical literacy are two dimensions of eHealth literacy, which reflect individuals’ basic and more advanced skills for using online health information, respectively. In this study, functional literacy was negatively associated with one’s addictive SMU. This finding was in line with the previous studies that found problematic SMU was more common among individuals with low health literacy [38], information literacy [47], or Internet literacy [39,48], and all these literacies comprise the basic skills of information reading and understanding, summarized as functional literacy. In addition, in the present study, functional literacy was a significant moderator in the association between perceived information distortion on social media and addictive SMU, and the moderating effect was only evident among those with lower functional literacy. Based on the Uses and Gratifications Theory [27], people’s social media use is directed by their need for gratification, and informative gratification can be impacted by the quality of informational content [29]. In the context of the COVID-19 pandemic, people use social media for more COVID-19 vaccine-related news to make health decisions (such as vaccination). Therefore, they need to obtain gratifying information on social media first. However, individuals with lower functional literacy generally have difficulty understanding new terms or knowledge about COVID-19 or vaccines [49]. When they perceive online news as being distorted, they may increase social media usage for more relevant information to familiarize themselves with such knowledge and reduce uncertainty or confusion, which may increase their risk of addictive SMU. In contrast, consumers with higher functional literacy can effectively capture and understand such information. Thus, they are less likely to get lost in distorted COVID-19 vaccination news or become addicted to social media.

In terms of critical literacy, the positively moderating role of critical literacy on the association between perceived information distortion was also observed in this study. Critical literacy involves the most advanced cognitive skills of eHealth literacy, reflecting one’s ability to critically evaluate the quality of online information and make proper decisions based on their judgments [50]. The importance of critical literacy has been emphasized in previous studies since the quality of online health information varies widely, especially on social media platforms. Several studies reported that the influence of critical literacy on health-promoting behaviors was higher than functional literacy, and individuals with high critical literacy might attempt to obtain as much information as possible and apply quality information to reach an optimal decision based on involvement theory [37,51,52,53]. The present study found that individuals with higher critical literacy were also vulnerable to developing addictive SMU when they perceived information distortion on social media. Past research has shown that people with higher critical literacy are generally inclined to use their rational knowledge to analyze the whole situation, retrieve more relevant information, and compare news from different channels when they use social media with informational motives [37,43,52,53], which in turn might also increase their risk of addictive SMU. Hence, although critical literacy is vital for people to evaluate information quality, its potential contribution to addictive SMU warrants attention.

A meta-analysis across 32 nations suggested that the prevalence of SMA was estimated as 25% [95% CI: 21–29%] for the studies adopting the same cutoff, and the prevalence was estimated to be 14% [95% CI: 9–19%] in individualist nations [54]. In the current sample of Hong Kong social media users, according to the cut-off value of 19 (out of 30) on the BSMAS, 67 (16.3%) had SMA, similar to the average pooled prevalence in individualist nations from the review. Some studies have reported an increase in the Internet or social media use with addictive tendencies after the COVID-19 outbreak. In addition, social distancing strategies were implemented to control the spread of COVID-19 worldwide. With these restrictions, the intensity of people’s use of social media may increase as it is the most accessible means of communication and obtaining news in real-time [11,12,55]. Therefore, addictive SMU can be a significant public health issue that should be watched out for amid the pandemic. For its potential predictors, previous research has limited evidence about the impact of information quality on one’s addictive SMU. This study fills this gap by examining the relationship between perceived information distortion regarding COVID-19 vaccination on social media and addictive SMU. In the present study, no significant association was directly found between perceived information distortion and addictive SMU, but the significant moderation roles of eHealth literacy could be observed, which probably be supported by past research that has shown individuals’ informational motives (e.g., keeping up with daily news) were not as strong predictors as entertainment motives of their addictive or problematic SMU [56]. Due to the limited evidence, the information-related predictors of addictive SMU and their underlying mechanisms still need further investigation.

### 4.1. Implications

Findings obtained from the present study have important implications for public health interventions to reduce the risk of addictive SMU among social media users during the COVID-19 pandemic. Findings provide clear evidence that individuals’ perceptions of information distortion regarding COVID-19 vaccination on social media, functional literacy, and critical literacy should be targeted for preventing addictive SMU or SMA among social media users. First, the important role of functional literacy documented in the present study suggests that individuals with lower functional literacy should be prioritized for interventions or health education to prevent social media addiction. Improving the functional literacy of social media users can be an effective approach to reducing their risk of addictive SMU during the pandemic. For example, education campaigns can be developed to improve the variety of search strategies, pandemic- or vaccine-related terms, and basic health knowledge for the public with actionable steps. In addition, some relevant directories and search engines should be developed to guide social media users to information related to COVID-19 vaccination [57].

Second, findings also suggest that higher critical literacy could concern social media users as they are particularly vulnerable to addictive SMU when they perceive COVID-19 vaccine-related news as being distorted on social media. Critical literacy is crucial for people to evaluate the information they receive and make informed decisions, which plays a vital role for users in preventing the adverse consequences caused by distorted information and better utilizing quality information on social media. However, with the extensive information on social media, special attention should also be given to individuals with high critical literacy to minimize their risk of addictive SMU.

Third, social media has become an ally but also a potential threat to public health amid the pandemic. With social media, information is shared in a faster way across geographical boundaries and time zones to address critical issues related to the pandemic [58]. However, the main concern of social media is its ability to disseminate a sheer amount of information quickly without formal peer review, which may confuse the public and increase their perceived information distortion on social media. Therefore, a joint industry effort should be made by media platforms against misinformation by elevating authoritative content and sharing critical updates coordinated by government healthcare agencies worldwide. Other interventions, including simplifying medical jargon and using easier figures or videos instead of words on social media, are advocated to reduce users’ perceived information distortion. Furthermore, social media users should also be encouraged to take responsibility for delivering the most trustworthy information during this period of uncertainty and squash-recognized misinformation.

### 4.2. Limitations

There were several limitations to the study that should be noted. First, the study was cross-sectional in nature, and thus causality cannot be inferred from these results. Second, some reporting bias (e.g., social desirability bias and recall bias) may exist as the questionnaire was self-reported and self-administered, although self-reported measures are more feasible to implement. Third, the sample was made up of social media users who voluntarily participated in this study. In other words, the participants were more likely to be interested in social media use or acquiring COVID-19-related news on social media. Therefore, the current sample might not represent the whole social media user population in Hong Kong. Nevertheless, we have strived to recruit participants with socio-demographic characteristics similar to the census data for adults aged 18 or over in Hong Kong [59]. Future studies using a population-representative sample are needed to further examine their relationships. Therefore, other information-related factors during the pandemic, such as information overload and information seeking, may also impact their addictive SMU behavior but were not included in the present study.

## 5. Conclusions

The present study identified the relationship between perceived information distortion, functional literacy, and critical literacy on addictive SMU. Results showed that lower functional literacy contributed significantly to addictive SMU among social media users; in addition, findings suggested that in individuals with lower functional literacy or higher critical literacy, those with higher perceived information distortion regarding COVID-19 vaccination on social media appear more likely to develop addictive SMU. Interventions to prevent addictive SMU in the COVID-19 pandemic should seek to improve the functional literacy of social media users; special attention should also be paid to the potential influence of higher critical literacy on addictive SMU. In addition, a joint effort should also be made by the media news agencies worldwide against distorted information and elevate authoritative content to improve the information quality on social media.

## Figures and Tables

**Figure 1 ijerph-19-08550-f001:**
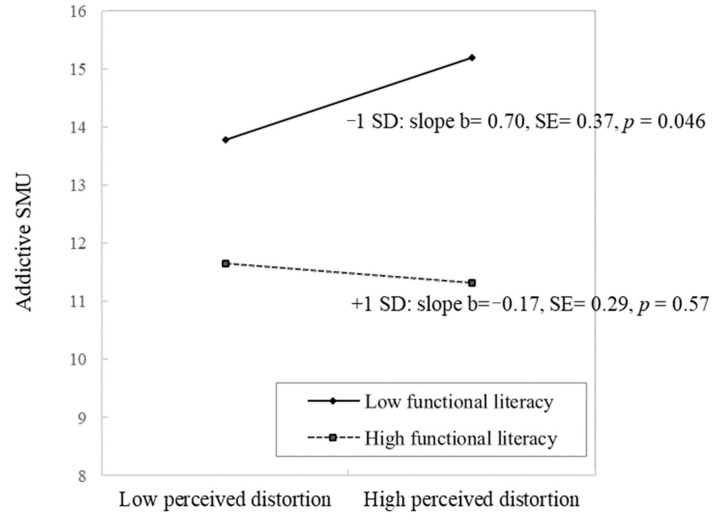
Interaction between perceived information distortion on social media and functional literacy on addictive SMU (N = 411).

**Figure 2 ijerph-19-08550-f002:**
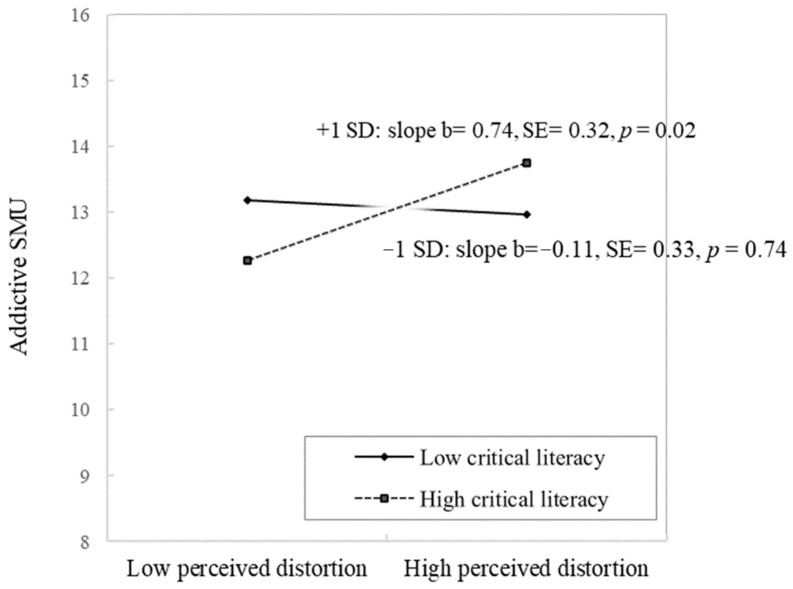
Interaction between perceived information distortion on social media and critical literacy on addictive SMU (N = 411).

**Table 1 ijerph-19-08550-t001:** Background characteristics of the participants (N = 411).

	N/Mean	(%/SD)
Age	34.54	15.30
Gender		
Male	169	41.1
Female	242	58.9
Marital status		
Single	263	64
Married	124	30.2
Cohabitating/divorced/widowed	24	5.8
Educational level		
Below university	87	21.2
University or above	324	78.8
Occupational status		
Full time	181	44.1
Student	137	33.3
Retired	38	9.2
Part-time/unemployed/housewife	55	13.4
Monthly income		
Below 10,000	186	45.3
10,000–19,999	68	16.6
20,000–39,999	100	24.3
40,000 above	54	13.1
Refuse to disclose	3	0.7
SMA	67	16.3

**Table 2 ijerph-19-08550-t002:** Correlation between variables (N = 411).

	1	2	3	4	5	6	7	8	9	10
1. Age	1.00									
2. Gender	0.01	1.00								
3. Marital Status	0.69 **	0.04	1.00							
4. Education	−0.51 **	−0.06	−0.45 **	1.00						
5. Employment	−0.07	0.06	0.04	−0.10 *	1.00					
6. Monthly income	0.48 **	−0.08	0.26 **	−0.07	−0.67 **	1.00				
7. Perceived information distortion	0.20 **	0.01	0.08	−0.06	0.09	0.02	1.00			
8. Functional literacy	−0.14 **	−0.07	−0.14 **	0.11 *	0.05	−0.08	−0.13 **	1.00		
9. Critical literacy	−0.11 *	0.01	−0.08	0.11 *	0.07	−0.11 *	0.01	0.08	1.00	
10. Addictive SMU	−0.23 **	0.04	−0.18 **	0.11 *	0.03	−0.14 **	0.02	−0.26 **	0.03	1.00

** *p* < 0.01, * *p* < 0.05.

**Table 3 ijerph-19-08550-t003:** Hierarchical regression analysis of perceived information distortion and functional literacy on addictive SMU (N = 411).

	Model 1		Model 2		Model 3	
	β	95% CI	β	95% CI	β	95% CI
*Block 1: Demographic variables*						
Age	−0.19 *	(−0.33, −0.04)	−0.23 **	(−0.37, −0.09)	−0.24 **	(−0.38, −0.10)
Education						
Below university	ref		ref		ref	
University or above	0.00	(−0.12, 0.11)	0.00	(−0.11, 0.11)	0.00	[−0.11, 0.11]
Marital status						
Single-divorced-cohabited	ref		ref		ref	
Married	−0.04	(−0.17, 0.08)	−0.04	(−0.16, 0.08)	−0.04	[−0.16, 0.08]
Monthly income						
10,000 or above	ref		ref		ref	
below 10,000	0.04	(−0.06, 0.14)	0.04	(−0.06, 0.13)	0.04	[−0.06, 0.13]
Incremental *R*^2^ (%)	5.30 ***		–		–	
*Block 2: Perceived information distortion and functional literacy*						
Perceived information distortion (IV)	–		0.03	(−0.06, 0.12)	0.05	(−0.04, 0.15)
Functional literacy (M1)	–		−0.29 ***	(−0.38, −0.20)	−0.29 ***	(−0.39, −0.20)
Incremental *R*^2^ (%)	–		8.60 ***		–	
*Block 3: Interaction*						
IV*M1	–		–		−0.10 *	(−0.19, 0.00)
Incremental *R*^2^ (%)	–		–		0.90 ***	
Total *R*^2^ (%)					14.80 ***	

* *p* < 0.05, ** *p* < 0.01, *** *p* < 0.001.

**Table 4 ijerph-19-08550-t004:** Hierarchical regression analysis of perceived information distortion and critical literacy on addictive SMU (N = 411).

	**Model 1**		**Model 2**		**Model 3**	
	**β**	**95% CI**	**β**	**95% CI**	**β**	**95% CI**
*Block 1: Demographic variables*						
Age	−0.19 *	(−0.33, −0.04)	−0.21 **	(−0.35, −0.06)	−0.21 **	(−0.35, −0.06)
Education						
Below university	ref		ref		ref	
University or above	0.00	(−0.12, 0.11)	−0.01	(−0.12, 0.11)	−0.01	(−0.12, 0.11)
Marital status						
Single-divorced-cohabited	ref		ref		ref	
Married	−0.04	(−0.17, 0.08)	−0.04	(−0.16, 0.09)	−0.04	(−0.17, 0.09)
Monthly income						
10,000 or above	ref		ref		ref	
Below 10,000	0.04	(−0.06, 0.14)	0.03	(−0.07, 0.13)	0.03	(−0.07, 0.13)
Incremental *R*^2^ (%)	5.30 ***		–		–	
*Block 2: Perceived information distortion and critical literacy*						
Perceived information distortion (IV)	–		0.06	(−0.03, 0.16)	0.06	(−0.04, 0.16)
Critical literacy (M2)	–		0.00	(−0.10, 0.09)	−0.01	(−0.10, 0.09)
Incremental *R^2^* (%)	–		0.40 **		–	
*Block 3: Interaction*						
IV × M2	–		–		0.10*	(0.00, 0.19)
Incremental *R*^2^ (%)	–		–		0.90 ***	
Total *R*^2^ (%)					6.60 ***	

* *p* < 0.05, ** *p*< 0.01, *** *p* < 0.001.

## Data Availability

The data presented in this study are available on request from the corresponding author. The data are not publicly available due to the privacy of participants.

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
