# Peer review of "Perceived Information Distortion about COVID-19 Vaccination and Addictive Social Media Use among Social Media Users in Hong Kong: The Moderating Roles of Functional Literacy and Critical Literacy"

_ijerph, 2022, doi:10.3390/ijerph19148550_

Round 1
Reviewer 1 Report
Although the article is coherent and its methodology is statistically well executed, the paper covers, perhaps, too many variables to be resolved with a survey that, in my opinion, is unrepresentative of the population studied. By not segmenting the sample, the generalisations made are not very concise about the sample surveyed. It is, in this sense, a poor contribution to the problem stated.
Also, digital literacy, both critical and functional, cannot be measured from a self-reported survey. Some alternative method of information verification is needed. How is critical assessment of information measured? Different studies by Unesco as well as the European Union have proposed whole yearbooks on how to measure it, and it is a very complex issue. It is the same with critical thinking: how has it been measured in the study? The survey questions do not define reality, but a perception, which, as pointed out, is a very important limitation of the article.
I insist that the universe is very large and therefore the sample is not representative of this universe.
At the same time, the survey as a factor of self-reported social media use has also been shown to be out of step with actual consumption data.
In the discussion of the results, some general assessments are made, which are not adequatesly qualified and, according to the results, cannot be generalised.
Likewise, the measurement of critical literacy and its relationship with SMU is not entirely clear or methodologically well worked out, basically because it is based on a survey and, from this, a correlation is declared which is difficult to understand. The first implication does not seem to be consistent with the text, since increasing critical literacy (at least self-reported in the survey) does not guarantee that there will be less addictive use of social media.
Overall, I believe that the proposed methodology does not resolve, first, the correlation and, second, the proper assessment of both social media use and critical skills related to media literacy.
Author Response
Thank you very much for your review of this manuscript.

Reviewer 2 Report
Dear Authors,
thank you for providing very interesting research. The topic you've focused on is important and up-do-date. I do not find any major mistakes or discrapancies between the stated research queastions and hypothesis and the conducted study. The structure of the article is well-organized and clear to the reader. The methods of the research were chosen approriatelly, so they are adequate to the stated research hypothesis.
However, I advise to:
a) in the Limitation section add information about potential further research that can be conducted in order to fill the gap - for instance, do you recommend research using the same method but focusing on different method of the recruitment for the research (propabilistic this time)? Is, in your opinion, 411 completed responses enough as the representative group for all the Hong Kong population as well as the Hong Kong social media users or maybe similar research should be conducted focusing on a larger group of participatns? Any addidtional paragraph in this particular section can be value in terms of pointing out the direction of further research.
b) in Conclusion section I recommend to emphasize to whom (SM users in general? governmental institutions holding a social media profile? news agencies sharing the health-related information?) you advise "Interventions to prevent addictive SMU in the COVID-19 pandemic should seek to improve the social media environment and individual’s functional literacy" (line 403-405)? In present form, in my opinion, it is not clear enough whose responsibility is it.
Author Response

(The authors gave the same response as above.)

Reviewer 3 Report
Well done. However, in Table 1, the sum of % should be 100, so plz reconsider them as the sum of Occupational status, and Monthly income not 100%.
Author Response

(The authors gave the same response as above.)

Reviewer 4 Report
I am very satisfied about the present manuscript. I commend the authors for their clarity.
I have just a minor issue. In Discussion the authors wrote that given the cross-sectional nature of the study, they were not able to make a causal prediction. However, regression could be considered a predictive analyses (compared to the correlations). There is a great debate about the possibility to do causal prediction in cross-sectional study. In my opinion a cross-sectional study can allow making also causal predictions depending on the order of the presentation of the scales. For this reason, I would not mention as limitation that the cross-sectional nature of study did not allow causal prediction, but maybe it is better to declare that the cross-sectional nature of the study did not allow hypotheses about mediation.
Authors have reported as a limitation also that the sample could not be representative. Please clarify better what you mean by this sentence. Of course, they have not used a probabilistic sample. At the same time, the sampling criteria seem sufficient to be confident about the generalizability of the sample.
Author Response

(The authors gave the same response as above.)

Round 2
Reviewer 1 Report
The given explanations for the suggested changes justify the decision taken in the article.